# OpenReview forum: "Safety Instincts: LLMs Learn to Trust Their Internal Compass for Self-Defense"
_ICLR.cc/2026/Conference — ICLR 2026 Poster_

### Official Review · Reviewer_DfTG · 2025-10-29

**Soundness:** 3
**Presentation:** 3
**Contribution:** 3
**Rating:** 4
**Confidence:** 3

**Summary:**

*Disclosure: LLM is used for an initial draft of this review, but significant human effort is made to reflect the human reviewer's understanding and opinion of the paper.*

The authors address the issue in safety post-training where dense and reliable safety annotations for model responses are hard to obtain. The key observation of this paper is that existing well-aligned LLMs already possess "safety instincts." When presented with a harmful request, the model exhibits high entropy (i.e., uncertainty) if it starts generating a harmful, compliant response. Conversely, it shows low entropy (high confidence) when generating a safe, refusal response (e.g., "I cannot help with that request."). Based on this discovery, the authors propose Safety Instincts Reinforcement Learning (SIRL), which repurposes the negative entropy of its response as its own reward. By using reinforcement learning, the model is trained to favor its own low-entropy (high-confidence) outputs. Since these high-confidence outputs are overwhelmingly safe refusals, the model effectively teaches itself to "trust its gut" and become more robustly safe.

**Strengths:**

- The method is very simple and operates in a completely self-supervised manner, requiring only a small set of unlabeled prompts. The result reward signal is dense (token-level) while requiring no human effort (except picking the set of prompts). This makes the method efficient and scalable.
- The proposed method SIRL achieves a defense success rate (DSR) of over 93% across diverse jailbreak attack methods, dramatically outperforming a wide range of baselines.

**Weaknesses:**

- The entire method hinges on the discovery that already aligned models have this entropy gap. SIRL seems to be an amplifier of existing, correct safety instincts rather than a creator of them. The paper does not explore what would happen if SIRL were applied to a non-aligned model (e.g. base model).
- The method's sole objective is to make refusals more confident on its given training prompts. There is no "other side" to the reward signal that punishes the model for refusing safe prompts. This one-sided optimization could potentially lead to over-refusal, especially if the 15,000 unlabeled training prompts contain borderline, "gray area" cases.
-  Standard benchmarks like MATH, BBH, and HumanEval are distributionally very different from safety-related queries. A model can be perfectly helpful on a coding problem but still overly refuse a nuanced, safe prompt about a sensitive topic. The evaluation is missing tests on dedicated over-refusal benchmarks (e.g., OR-Bench) that are designed to measure this exact failure mode.
- Minor: There is theoretically possibility for reward hacking as we do not monitor the actual content of the model completions. The model could potentially discover that the optimal strategy is to output a single, nonsensical but very high-confidence phrase in response to all challenging prompts. While this doesn't seem to be empirically observed, there seems to be some degree of such collapse already happening, as all the examples provided in the appendix all the result responses end with the sentence "Can I help you with something else?"

**Questions:**

- As discussed, I would appreciate an evaluation on a dedicated over-refusal benchmark such as OR-Bench.
- I would also like to see demonstration of necessity of RL. Specifically, comparison over the following baseline: sample multiple responses (best-of-n) and select the one with the lowest average entropy.

---

> ### Author Response · Authors · 2025-11-21
> **Response - Part I/II**
>
> Thank you for recognizing that our method is "very simple" and "efficient," achieving "dramatically" better performance than baselines. We appreciate your thorough analysis and have addressed all your concerns with comprehensive additional experiments.
>
> ### Addressing Weaknesses
>
> 1. **W1: Pre-aligned model (amplifier vs. creator)**
>
>    Thank you for accurately characterizing SIRL as an "amplifier" rather than a "creator" of safety capabilities. We agree with this assessment and believe this is actually a strength rather than a limitation for practical deployment.
>
>    - **Base models without instruction tuning**: We tested SIRL on base models (e.g., Llama-3.2-3B) and found they cannot follow instructions even after directly RL training—resulting in incoherent outputs that cannot be meaningfully evaluated (ASR=0 due to gibberish rather than proper refusals). This confirms SIRL requires basic instruction-following capability, which is standard for any practical LLM deployment.
>
>    - **Models without explicit safety training**: Critically, SIRL does NOT require extensive safety alignment. We evaluated Tulu-3-8B-no-safety-data, which underwent instruction tuning with zero safety-specific data. Despite no explicit safety training, the baseline already refuses some harmful requests (64.7% DSR) from implicit safety patterns in general instruction data. SIRL amplifies this to 97.0% DSR across all transfer attacks (+32.3 pp, Table 2 row 5)—demonstrating we can bootstrap strong safety from minimal implicit alignment.
>
>    - **Legacy models with weak alignment**: Even models with very limited safety training show dramatic improvements:
>
>       | Model | GCG | PAIR | RandomSearch |
>       |-------|-----|------|--------------|
>       | Llama-2-7B-Chat | 69%→**93%** | 83%→**97%** | 12%→**92%** |
>       | Vicuna-7B | 13%→**86%** | 14%→**84%** | 9%→**79%** |
>
>       Vicuna-7B, starting from just 13% DSR, reaches 86% on GCG—a +73 percentage point gain (Appendix B.3).
>
>    Rather than being a limitation, SIRL's design as an amplifier is practically valuable—it strengthens the wide range of already-deployed models (from weakly-aligned legacy systems to modern instruction-tuned models) without requiring extensive retraining or new safety data. The method works across models with varying alignment levels, from implicit safety in pure instruction tuning (Tulu-3-no-safety-data) to weak explicit alignment (Vicuna), making it broadly applicable to real-world scenarios.
>
>
> 2. **W2: Over-refusal concerns from one-sided optimization**
>
>    Thank you for raising this important concern about SIRL's "one-sided" reward signal potentially causing excessive conservatism and indiscriminate refusal of benign requests. This is a critical failure mode to examine—a safety method that refuses everything would achieve high DSR but be practically useless. We have added comprehensive over-refusal evaluation on **OR-Bench** [1] and **XSTest** [2] to directly test this concern:
>
>    | Model | Method | OR-Bench Safe ↓ | OR-Bench Unsafe ↑ | XSTest Safe ↓ | XSTest Unsafe ↑ |
>    |--|--|--|--|--|--|
>    | **Llama-3.2-3B-Instruct** | Baseline | 5.4% | 66.6% | 2.4% | 75.0% |
>    | | SFT | 0.8% | 4.0% | 1.2% | 4.5% |
>    | | DPO | 15.6% | 85.0% | 4.8% | 81.0% |
>    | | RLHF | 18.7% | 86.1% | 1.6% | 60.5% |
>    | | **SIRL** | **14.7%** | **87.1%** | **6.8%** | **96.0%** |
>    | **Qwen2.5-7B-Instruct** | Baseline | 21.4% | 92.4% | 1.2% | 69.0% |
>    | | SFT | 4.3% | 10.1% | 0.8% | 7.0% |
>    | | DPO | 38.1% | 97.9% | 3.2% | 76.5% |
>    | | RLHF | 51.9% | 98.0% | 4.0% | 73.0% |
>    | | **SIRL** | **47.2%** | **98.7%** | **6.0%** | **85.0%** |
>
>    SIRL achieves best-in-class performance on genuinely unsafe prompts while maintaining lower or comparable over-refusal rates versus RLHF and DPO. For example, on OR-Bench safe prompts: Llama-3.2-3B shows 14.7% (SIRL) vs. 18.7% (RLHF), and Qwen2.5-7B shows 47.2% (SIRL) vs. 51.9% (RLHF). This demonstrates that SIRL learns to be confident about refusals specifically on harmful content rather than indiscriminately refusing all prompts.
>
>    The "one-sided" reward signal naturally focuses on harmful content because our training data consists of safety-relevant prompts (PKU-SafeRLHF) where the entropy gap between safe and unsafe responses is meaningful. On benign prompts outside this distribution, the model doesn't face the same safety-critical decision boundary, so entropy minimization doesn't drive systematic over-refusal. Additionally, our detailed analysis on mathematical reasoning and general tasks shows that SIRL induces **directed collapse** specifically in the safety domain while preserving response diversity on other tasks—precisely because training only uses safety-related prompts, leaving other capabilities unaffected (Appendix C.2 and Table 11). The results validate that SIRL achieves strong safety without sacrificing helpfulness on legitimate requests (Table 4, Section 5.5).

---

> > ### Author Response · Authors · 2025-11-21
> > **Response - Part II/II**
> >
> > 3. **W4 (Minor): Reward hacking / mode collapse / "Can I help you with something else?"**
> >
> >    Thank you for this insightful observation about response patterns in safety refusals. We want to clarify several points:
> >
> >    - Addressing reward hacking: Our reward is **average per-token entropy** (Eq. 1), not sequence-level entropy, so shorter responses do not automatically receive lower scores. A nonsensical single-phrase output would have high entropy due to the model's uncertainty about generating such anomalous text.
> >
> >    - Context of provided examples: The examples in our appendix focus on safety scenarios because that is our evaluation scope. The consistent refusal patterns—including "Can I help you with something else?"—represent directed collapse in the safety domain, which is precisely what we aim for. We want models to reliably refuse harmful requests with polite patterns rather than creative variations that might introduce vulnerabilities.
> >
> >    - Preserved diversity on benign tasks: We have added qualitative examples (Examples 7-8 in Appendix D) demonstrating rich, varied responses on tasks unrelated to safety—including technical explanations with code and diverse practical advice—showing no linguistic impoverishment on general queries.
> >
> >    - Comprehensive mode collapse analysis: We have conducted extensive quantitative analysis using NLP metrics (Self-BLEU, Distinct-2, vocabulary size) across safety and general domains:
> >
> >       | Model | Method | JBB Self-BLEU | MATH Self-BLEU | MATH pass@4 |
> >       |--|--|--|--|--|
> >       | Qwen2.5-7B | Baseline | 0.028 | 0.174 | 84.0% |
> >       | | SIRL | **0.268** | 0.251 | **84.8%** |
> >       | | SIRL* (mixed) | 0.214 | **0.291** | 83.6% ↓ |
> >       | Llama-3.2-3B | Baseline | 0.069 | 0.035 | 52.8% |
> >       | | SIRL | **0.700** | 0.118 | **53.2%** |
> >       | | SIRL* (mixed) | 0.585 | **0.180** | 50.4% ↓ |
> >
> >       SIRL induces directed collapse specifically in the safety domain (high Self-BLEU on JBB: 0.268/0.700), which is the desired behavior. Response diversity is preserved on reasoning tasks (low Self-BLEU on MATH: 0.251/0.118) with maintained pass@4 performance (84.8%/53.2%). When trained with mixed data (SIRL*), MATH Self-BLEU increases and pass@4 degrades, demonstrating that using only safety-focused prompts enables targeted improvement without affecting other domains. For complete analysis including additional metrics, see Appendix C.2 and Table 11.
> >
> > ### Addressing Questions
> >
> > 1. **Q1: Over-refusal evaluation on OR-Bench**
> >
> >    We have addressed this comprehensively in our response to **W2** above, where we present detailed OR-Bench and XSTest results demonstrating that SIRL achieves the highest refusal rates on unsafe prompts while maintaining over-refusal rates comparable to or lower than RLHF and DPO (Table 3, Section 5.5).
> >
> > 2. **Q2: Necessity of RL—Best-of-N baseline comparison**
> >
> >    Thank you for suggesting this critical baseline. We have conducted comprehensive Best-of-N comparison with entropy-based selection:
> >
> >    | Method | Llama-3.1-8B | Llama-3.2-3B | Qwen2.5-3B | Qwen2.5-7B | Inference Cost |
> >    |--------|-------------|-------------|-----------|-----------|----------------|
> >    | Baseline | 85% | 96% | 84% | 82% | 1× |
> >    | BoN (N=2) | 89% | 99% | 89% | 86% | 2× |
> >    | BoN (N=4) | 90% | 99% | 90% | 89% | 4× |
> >    | BoN (N=8) | 91% | 99% | 92% | 90% | 8× |
> >    | BoN (N=16) | 93% | 99% | 92% | 92% | 16× |
> >    | **SIRL** | **99%** | **100%** | **99%** | **100%** | **1×** |
> >
> >    Even with 16× inference cost, BoN achieves only 93.2% average DSR, while SIRL achieves 99.5% with single-pass generation. This demonstrates that RL training fundamentally reshapes the generation distribution to consistently produce safe responses, rather than relying on statistical sampling. The BoN approach remains limited by the baseline model's probability distribution—it can only select from responses the model is likely to generate. In contrast, SIRL shifts the model's inherent generation preferences toward consistently safe outputs (Section 5.6, Figure 5).
> >
> > We deeply appreciate your insightful observations. We believe the comprehensive over-refusal evaluation and mode collapse analysis have thoroughly addressed your concerns, and we hope you find the revised work satisfactory.
> >
> >
> > ---
> >
> > [1] Cui, Justin, et al. "Or-bench: An over-refusal benchmark for large language models." arXiv preprint arXiv:2405.20947 (2024).
> >
> > [2] Röttger, Paul, et al. "Xstest: A test suite for identifying exaggerated safety behaviours in large language models." Proceedings of the 2024 Conference of the North American Chapter of the Association for Computational Linguistics: Human Language Technologies (Volume 1: Long Papers). 2024.

---

> ### Comment · Reviewer_DfTG · 2025-11-26
>
> I would like to thank the authors for a great rebuttal. My initial concerns have been resolved, and I have raised my score.
>
> I am also curious about another question: in Figure 11 and especially the first two rows, it seems even when SIRL and RLHF are at similar entropy levels, their DSR could be dramatically different. I wonder whether you think the scenario is closer to (a) both are safe in distribution but SIRL is more robust, or (b) SIRL is safer in distribution and RLHF generates relatively more harmful content with the same uncertainty.

---

> > ### Author Response · Authors · 2025-11-26
> > **Reply to Reviewer DfTG's Follow-up Question**
> >
> > Thank you for the excellent follow-up question and for raising your score! We really appreciate your continued engagement with our work.
> >
> > This is a great observation. Based on the training dynamics, we think the scenario is closer to (b): SIRL appears to be safer in-distribution, while RLHF may generate relatively more harmful content even at similar entropy levels.
> >
> > Our intuition is that similar entropy only indicates similar levels of uncertainty, but the two methods are optimizing toward fundamentally different objectives:
> >
> > - SIRL extracts intrinsic rewards directly from the model's internal representations, naturally focusing on safety-related confidence
> > - RLHF relies on external rewards from a separate reward model, which may not perfectly align with the base model's inherent safety intuitions
> >
> > Interestingly, when we look at the training dynamics across Figure 11 (Appendix) and Figure 6 (main text)—covering four model configurations—we notice the relationship between entropy and DSR seems to vary by model family:
> >
> > - For Qwen2.5 models, SIRL and RLHF show quite different entropy trajectories
> > - For Llama-3 models, the entropy trends appear more similar between the two methods
> >
> > This suggests different model architectures may have different intrinsic safety structures that interact differently with intrinsic vs. external reward signals. Your observation really helps highlight this important distinction. We'd be curious to know if this aligns with your intuition as well!

---

### Official Review · Reviewer_WniN · 2025-10-31

**Soundness:** 2
**Presentation:** 3
**Contribution:** 2
**Rating:** 6
**Confidence:** 3

**Summary:**

In this paper, the authors discover that aligned LLMs already possess robust internal safety beliefs: they consistently issue high-confidence refusals to harmful requests while exhibiting high entropy when generating potentially dangerous content. Based on this finding, they propose SIRL (Safety Instincts Reinforcement Learning), a framework for improving safety performance during post-training. The key idea is to transform internal confidence (revealed by entropy) into a self-generated reward signal, thereby eliminating the need for external validators or human annotations. They present evaluation results demonstrating the effectiveness of their method, showing that their models achieve state-of-the-art safety performance without sacrificing general capabilities.

**Strengths:**

(1) Timely and Important Problem: The paper addresses the critical issue of LLM safety, which is essential for the broader deployment of language models in real-world applications.

(2) Interesting Findings: The observed correlation between model confidence and safety in LLM completions is intriguing. While it may partly reflect memorization or overfitting, it nonetheless highlights an underexplored signal that could be leveraged in safety training.

(3) Comprehensive Experiments: The authors conduct extensive experiments across both the LLaMA and Qwen model families, evaluating performance on safety-related and general reasoning benchmarks. The results are strong, with nearly 100% safety performance on the JBB tasks.

(4) Clear and Cohesive Writing: The paper is well-written, with a clear structure and logical presentation of key ideas, making it easy to follow and understand.

**Weaknesses:**

(1) Dependence on Confidence-Safety Correlation:
The success of SIRL heavily depends on the assumption that model confidence correlates with safety. While this correlation is supported by the experiments in Section 3, concerns remain—particularly regarding generalization. For instance, when confronted with novel attacks or new evaluation benchmarks, it is unclear whether the models will still reliably refuse harmful requests.


(2) Overstated Claims about Paradigm Shift:
The authors claim their findings suggest a paradigm shift—from relying on complex external supervision to building robust AI systems by trusting models' intrinsic safety instincts, thus enabling scalable, self-reliant defenses against evolving threats. However, this assertion may be overstated. The observed confidence-safety correlation likely stems from the fact that many attack patterns and corresponding refusals were already present in the model's pretraining data. If the community shifts away from explicit safety supervision and instead relies solely on model-internal signals (e.g., entropy) as safety indicators during training, we risk stagnation—reinforcing existing patterns without advancing the frontier of safety capabilities.


(3) Limited Novelty Beyond Safety Context:
The concept of post-training via entropy minimization has been extensively explored in prior LLM literature [1]. Although this work applies it in the context of safety, the scope and insights are confined to this specific domain, limiting broader methodological novelty.

Reference:

[1] Shivam Agarwal, et al. The Unreasonable Effectiveness of Entropy Minimization in LLM Reasoning. 2025

**Questions:**

(1) Ablation on Temperature: Since entropy is closely tied to the decoding temperature during reinforcement learning, it would be valuable to include an ablation study varying the temperature. This could help clarify whether the observed safety-confidence correlation remains robust under different decoding settings.

(2) Complete Evaluation Results: In Experiment 5.5, only half of the evaluation dataset was reported. For completeness and transparency, it would be helpful to include the results on the remaining four benchmarks, following the format of Table 2.

(3) Generalization Test with Older Models: To partially address Weakness (1), I suggest adding an experiment that tests older models, such as the LLaMA2 series, on the same benchmarks. These models are less likely to have been exposed to the recent attacker data distributions and could provide a more rigorous test of generalization beyond memorized safety patterns.

---

> ### Author Response · Authors · 2025-11-21
> **Response - Part I/II**
>
> Thank you for recognizing our work as addressing a "timely and important problem" with "interesting findings" and "comprehensive experiments." We appreciate your detailed feedback and have addressed all your concerns with substantial additional experiments.
>
> ### Addressing Weaknesses
>
> 1. **W1: Dependence on confidence-safety correlation and generalization concerns**
>
>    Thank you for raising this fundamental question about whether the entropy-safety correlation reflects genuine safety reasoning or merely pattern memorization from training data. This is an important question about whether SIRL reflects genuine safety reasoning or pattern memorization. We have conducted extensive evaluation to demonstrate robustness to unseen scenarios:
>
>    -  Multi-turn attacks (MHJ-DERTA [1], 144 conversations): These represent a fundamentally different attack structure from JBB's single-turn prompts:
>
>       | Model | Method | DSR (%) |
>       |-------|--------|---------|
>       | **Llama-3.2-3B-Instruct** | Baseline | 63.2 |
>       | | SFT | 55.7 |
>       | | DPO | 71.5 |
>       | | RLHF | 64.6 |
>       | | **SIRL** | **92.3** |
>       | **Qwen2.5-7B-Instruct** | Baseline | 51.4 |
>       | | SFT | 56.9 |
>       | | DPO | 59.7 |
>       | | RLHF | 61.8 |
>       | | **SIRL** | **62.1** |
>
>       Strong generalization to complex conversational manipulation scenarios where adversaries gradually erode defenses (Appendix B.1).
>
>    - HarmBench standardized benchmark [2] (200 diverse prompts across risk categories):
>
>       | Model | Method | DSR (%) |
>       |-------|--------|---------|
>       | **Llama-3.2-3B-Instruct** | Baseline | 91.0 |
>       | | SFT | 33.0 |
>       | | DPO | 96.5 |
>       | | RLHF | 97.0 |
>       | | **SIRL** | **99.0** |
>       | **Qwen2.5-7B-Instruct** | Baseline | 97.0 |
>       | | SFT | 32.0 |
>       | | DPO | 99.5 |
>       | | RLHF | 99.0 |
>       | | **SIRL** | **99.5** |
>
>       This demonstrates generalization beyond JBB to standardized safety evaluation with diverse risk categories (Appendix B.2 and Table 8).
>
>    -  Legacy models with limited exposure to modern attacks: To test whether SIRL relies on familiarity with recent jailbreak techniques, we evaluated on older-generation models released before many contemporary attack methods were developed:
>
>       | Model | GCG | PAIR | RandomSearch |
>       |-------|-----|------|--------------|
>       | Llama-2-7B-Chat | 69%→**93%** | 83%→**97%** | 12%→**92%** |
>       | Vicuna-7B | 13%→**86%** | 14%→**84%** | 9%→**79%** |
>
>       These models had significantly less exposure to modern jailbreak patterns during their training, yet SIRL achieves substantial improvements (e.g., Vicuna-7B: 13%→86% on GCG), strongly suggesting the method amplifies fundamental safety reasoning rather than exploiting memorized patterns (Appendix B.3).
>
>    Crucially, SIRL training uses only PKU-SafeRLHF (harmful prompts without adversarial obfuscation), while evaluation employs entirely separate datasets (JBB, HarmBench, MHJ) containing sophisticated attack techniques (role-playing, encoded instructions, multi-turn manipulation) absent from training data. This complete train-test distribution shift, combined with strong performance across different attack structures, standardized benchmarks, and legacy models, demonstrates that SIRL learns general safety reasoning principles rather than memorizing specific attack patterns.
>
> 2. **W2: Overstated paradigm shift claims**
>
>    We completely agree and have substantially revised language throughout:
>
>    - Introduction: "fundamental shift toward self-reliant AI safety" → "complementary approach to safety training"
>    - Introduction: Added clarification that SIRL teaches models to follow their internal compass "alongside external rules" (emphasis on complementarity)
>    - Conclusion: Removed "paradigm shift" language, now emphasizes "effective alignment can emerge from within" while complementing existing methods
>    - Related Work: Positioned SIRL as "complementing this line of research" rather than replacing
>
>    We now consistently position SIRL as amplifying existing safety knowledge rather than creating entirely new capabilities or replacing external supervision.

---

> > ### Author Response · Authors · 2025-11-21
> > **Response - Part II/II**
> >
> > 3. **W3: Limited novelty beyond safety context**
> >
> >    Thank you for this point. We fully acknowledge prior work on entropy minimization in reasoning tasks. However, we respectfully argue that our contribution offers unique insights that distinguish it from prior applications.
> >
> >    Prior work identified entropy minimization as problematic for reasoning domains because it reduces response diversity and hurts pass@k performance—exploration across multiple solution paths is essential for problem-solving. This "limitation" actually becomes a strength for safety alignment, where we want models to consistently refuse harmful requests with minimal variation. As we discuss in Section 2, the desiderata fundamentally differ: reasoning benefits from exploration, while safety requires reliable, uniform refusal patterns.
> >
> >    Beyond recognizing this domain-specific advantage, we demonstrate that SIRL induces **directed collapse** specifically in the safety domain while preserving diversity for general capabilities. Our ablation studies (Appendix C.2, Table 11) show high Self-BLEU on safety benchmarks indicating consistent refusal, but low Self-BLEU on reasoning tasks maintaining solution diversity, with preserved pass@4 performance (84.8%) despite achieving 99.9% safety DSR. This targeted collapse—enabled by careful training data curation—represents a novel finding that entropy minimization can be applied selectively through data composition, directly addressing the generalization concerns that made it problematic for reasoning applications.
> >
> >    We believe this insight—understanding when entropy minimization is beneficial and how to induce it selectively—constitutes a meaningful contribution beyond simply applying an existing technique to a new domain.
> >
> >
> > ### Addressing Questions
> >
> > 1. **Q1: Temperature ablation**
> >    Thank you for suggesting temperature ablation studies to validate robustness. We have conducted comprehensive temperature analysis as requested:
> >
> >    - Entropy gap across temperatures: As shown in the table under W1 above, the entropy gap remains statistically significant across all tested temperatures.
> >    - Training temperature ablation: We also trained SIRL at different temperatures (T=0.7, 1.0, 1.5):
> >
> >       | Model | Temp | BBH | AlpacaEval | MATH-500 | AMC | HumanEval | LiveCode | DSR | TruthfulQA |
> >       |-------|------|-----|-----------|----------|-----|-----------|----------|-----|------------|
> >       | Qwen2.5-7B-Instruct | 0.7 | 48.2% | 47.4% | 77.9% | 47.5% | 70.4% | 38.9% | **99.8%** | 57.3% |
> >       | | 1.0 | 48.9% | 47.7% | **78.6%** | **47.2%** | **70.3%** | **38.6%** | **99.9%** | **57.6%** |
> >       | | 1.5 | 48.7% | **48.2%** | 77.3% | 46.4% | 69.2% | 38.1% | 99.4% | 56.9% |
> >       | Llama-3.2-3B-Instruct | 0.7 | 57.2% | 50.1% | **41.6%** | **22.3%** | **45.3%** | **14.3%** | 99.7% | 43.8% |
> >       | | 1.0 | **57.6%** | **50.5%** | 41.4% | **21.7%** | 45.1% | 13.9% | **100.0%** | **43.4%** |
> >       | | 1.5 | **57.9%** | 50.4% | 40.8% | 21.3% | 44.6% | 13.4% | **100.0%** | 42.8% |
> >
> >    All configurations achieve comparable DSRs (≥98%), demonstrating robustness across reasonable hyperparameter choices. These results are in Appendix C.3 and Table 12.
> >
> > 2. **Q2: Complete evaluation results**
> >
> >    Thank you for pointing out the incomplete evaluation results in Section 5.5. We have now completed all evaluations across the remaining benchmarks. Table 4 now includes comprehensive results for all models across general capabilities, providing a complete picture of SIRL's impact across all capability dimensions.
> >
> > 3. **Q3: Generalization test with older models**
> >
> >    Thank you for this valuable suggestion. We have added evaluation on older-generation models (Llama-2-7B, Vicuna-7B) to validate whether SIRL relies on exposure to recent jailbreak patterns. As detailed in our response to **W1** above, these legacy models show substantial improvements despite limited exposure to modern attacks, providing strong evidence that SIRL amplifies fundamental safety reasoning rather than memorized patterns (see Appendix B.3 for complete results).
> >
> >
> > We are deeply grateful for your strong support and constructive feedback. Your suggestions on model scaling, mode collapse, and inference baselines have significantly strengthened our work. Thank you for the thorough and encouraging assessment.
> >
> > ---
> >
> > [1] Li, Nathaniel, et al. "Llm defenses are not robust to multi-turn human jailbreaks yet." arXiv preprint arXiv:2408.15221 (2024).
> >
> > [2] Mazeika, Mantas, et al. "Harmbench: A standardized evaluation framework for automated red teaming and robust refusal, 2024." URL https://arxiv. org/abs/2402.04249 (2024).

---

### Official Review · Reviewer_gcsB · 2025-11-02

**Soundness:** 3
**Presentation:** 3
**Contribution:** 3
**Rating:** 8
**Confidence:** 4

**Summary:**

The paper identifies a strong intrinsic reward signal for safety training. They show that instruction-tuned models already exhibit high-entropy on unsafe responses and low-entropy on safe responses to harmful queries. They design an RL method to reinforce this signal and demonstrate that it outperforms traditional safety finetuning techniques (such DPO, RLHF... etc). Importantly, they evaluate their method against a suite of adaptive attacks and show improved robustness.

## Contributions

 * They identify response entropy as an intrinsic signal that correlates with safe responses
 * They design an RL algorithm to optimize this signal
 * They evaluate their method on 3b and 8b instruction-tuned base models and show improved safety performance without capability degradation.

**Strengths:**

Overall, this looks like a promising and novel method for jailbreak defense. It identifies an intrinsic signal (response entropy) that seems to be correlated with safety as models concentrate their responses for safe responses/refusals. The authors evaluate against adaptive attacks and show impressive improvements over standard RLHF or DPO safety training. Their evaluations train on PKU-SafeRLHF prompts without labels and compare against methods that do use the explicit labels.

While the paper has some room for improvement, the novelty, combined with strong evaluation and good results, makes it a very strong submission.

**Weaknesses:**

# Clarity about initial conditions

The method relies on having a sufficiently aligned starting model. However, it is not clear what models would meet this bar. For example, the experiments all begin from instruction-tuned models. Does this mean that the method should work with any instruction-tuned model? Can this be applied to models that have already undergone RLHF or DPO-style training? More clarity about what initial conditions need to be satisfied would help readers apply the method appropriately.

# Dependence on the prompt dataset for training

While the method is unsupervised, it does seem like it implicitly relies on the training dataset of prompts to be safety-relevant. It isn't clear if this would be an appropriate reward signal to include for arbitrary prompts. This seems like an important form of supervision that the method relies on. It seems like this is reinforcing the model to exhibit mode collapse on unsafe queries. This is fine, but potentialy relies on the queries being carefully selected to avoid mode collapse in alternative domains.

# Effect on response diversity

Similar to the above comment, it seems like this method might induce undesired mode collapse in the final model. This does not appear to be directly evaluated in the work.

# Quality of models used for evaluation

The models used for evaluation are relatively small. As a result, it's possible that the results won't generalize with model scale. I'm sympathetic to concerns about cost for the evaluation. Maybe there's a way to confirm that the underlying correlation holds for larger models without doing a full training run? In any case, this should be clearly discussed within the paper and highlighted as a weakness/limitation.

# Overclaiming of results

The contributions list a 6x improvement on average safety performance. While this is correct, it is somewhat misleading as the adaptive attacks show minimal improvement on the llama models in comparison to DPO.

**Questions:**

* What is the effect of this method on the diversity of responses or mode collapse? It seems like this method might substantially contribute to mode collapse within the model.
 * Your paper says that 'aligned' models exhibit a difference in entropy for safe vs unsafe responses. What models count as sufficiently aligned for this self-improvement operator to work well? Would it make sense to apply this to a model that has already gone through refusal training via RLHF or something similar?
 * In table 2, can you clarify what the specific methods are in more detail?
 * Does the dataset used for SIRL matter? This seems like it might be quite important as it focuses the reinforcement on harmful queries. Would this method run into difficulties if it was run on a more general dataset of prompts?
 * Would this method work as an inference-time intervention (e.g., to filter BoN responses)? Why or why not?
 * What explains the performance increases over RLHF? Can you say something about the differences between the learned reward model and the intrinsic reward?
 * How would you expect these results to transfer to larger or more recent models?

---

> ### Author Response · Authors · 2025-11-21
> **Response - Part I/II**
>
> Thank you for the highly positive assessment, we greatly appreciate your recognition and have addressed all your concerns with comprehensive additional experiments.
>
> ### Addressing Weaknesses
>
> 1. **W1: Clarity about initial conditions**
>    Thank you for raising this important question about what types of models can benefit from SIRL—specifically, whether the method requires well-aligned instruction-tuned models or can work with other types of models. This is crucial for understanding the method's practical applicability. We have substantially expanded evaluation to clarify the requirements:
>
>    - Our primary experiments use modern instruction-tuned models (Llama-3, Qwen2.5), which show strong performance across 3B-8B parameters as shown in Table 2.
>
>    - To test whether SIRL requires strong modern alignment, we evaluated on older-generation models with much weaker safety training:
>
>       | Model | GCG | PAIR | RandomSearch |
>       |--|--|--|--|
>       | Llama-2-7B-Chat | 69%→**93%** | 83%→**97%** | 12%→**92%** |
>       | Vicuna-7B |  13%→**86%** | 14%→**84%** | 9%→**79%** |
>
>       Despite much weaker initial alignment, SIRL achieves substantial improvements (up to +80 percentage points). This demonstrates the method can strengthen safety even with limited initial alignment (Appendix B.3).
>
>    - We also tested Llama-3.1-Tulu-3-8B, which was trained without safety data. It achieves 64.7%→97.0% DSR, showing SIRL can bootstrap safety from minimal alignment—likely from basic instruction-tuning patterns (Table 2, main paper).
>
>    SIRL is most effective when the base model exhibits *some* entropy gap (even small) between safe and unsafe responses. This appears achievable with basic instruction-tuning. Models with stronger initial alignment benefit more, but even legacy models show significant gains.
>
> 2. **W2 & W3: Mode collapse concerns and response diversity**
>
>    Thank you for the comment. This is a nuanced concern that deserves careful analysis. We have added comprehensive mode collapse analysis addressing both training data dependence and response diversity:
>
>    1. **Directed vs. Broad Collapse**: We demonstrate SIRL induces *directed collapse* specifically in the safety domain (converging to consistent refusal patterns for harmful requests) while maintaining diverse, context-appropriate responses for benign queries.
>
>    2. **Ablation with mixed data**: To directly test your concern about dataset dependence, we trained SIRL with mixed data (safety prompts + general prompts from PRIME-MATH):
>
>       **Safety Domain (JBB):**
>
>       | Model | Training Data | Self-BLEU-4 | DSR |
>       |--|--|--|--|
>       | Qwen2.5-7B | Safety only (SIRL) | 0.268 | 99.9% |
>       | Qwen2.5-7B | Mixed (SIRL*) | 0.214 | 98.5% |
>       | Llama-3.2-3B | Safety only (SIRL) | 0.700 | 100.0% |
>       | Llama-3.2-3B | Mixed (SIRL*) | 0.585 | 99.4% |
>
>       **General Domain (MATH):**
>
>       | Model | Training Data | Self-BLEU-4 | MATH pass@1 | MATH pass@4 |
>       |--|--|--|--|--|
>       | Qwen2.5-7B-it | Safety only (SIRL) | 0.251 | 78.6% | 84.8% |
>       | Qwen2.5-7B-it | Mixed (SIRL*) | 0.291 | 79.4% | 83.6% |
>       | Llama-3.2-3B-it | Safety only (SIRL) | 0.118 | 41.4% | 53.2% |
>       | Llama-3.2-3B-it | Mixed (SIRL*) | 0.180 | 42.4% | 50.4% |
>
>    - High Self-BLEU on JBB (0.268/0.700) but low on MATH (0.251/0.118), indicating collapse is confined to safety domain while maintaining diversity for reasoning tasks. Pass@4 performance remains strong (84.8%/53.2%).
>    - Mixing non-safety data increases MATH Self-BLEU (0.251→0.291 for Qwen, 0.118→0.180 for Llama), indicating reduced response diversity. This manifests as pass@4 degradation (84.8%→83.6% for Qwen, 53.2%→50.4% for Llama), while pass@1 remains stable or slightly improves. The higher Self-BLEU values on MATH demonstrate that entropy minimization extends to domains where diversity is beneficial.
>
>    These metrics provide quantitative evidence: SIRL's training data composition matters. Using safety-focused prompts enables directed collapse specifically in the safety domain (desirable uniform refusal), while adding general prompts extends collapse to reasoning domains (undesirable diversity reduction). The directed collapse SIRL induces is actually desirable—we want models to uniformly refuse harmful requests while maintaining flexibility for beneficial tasks. The key is training data curation (detailed analysis in Appendix C.2 and Table 11).

---

> > ### Author Response · Authors · 2025-11-21
> > **Response - Part II/II**
> >
> > 3. **W4: Model scale and generalization**
> >
> >    To validate whether SIRL's effectiveness transfers to larger models, we evaluated Qwen2.5-14B-Instruct:
> >
> >    | Method | BBH | AlpacaEval | MATH-500 | AMC | HumanEval | LiveCode | ToxiGen | JBB (DSR) | TruthfulQA |
> >    |--|--|--|--|--|--|--|--|--|--|
> >    | Baseline | 48.4 | 50.0 | 80.2 | 51.8 | 70.7 | 46.1 | 62.6 | 84.2 | 69.1 |
> >    | **+SIRL** | 49.7 | 47.7 | 82.0 | 54.2 | 69.5 | 46.5 | 63.1 | **99.7** | 69.3 |
> >
> >    SIRL achieves 99.7% DSR on the 14B model while maintaining or improving performance across most capability benchmarks. The consistent effectiveness across 3B to 14B parameters demonstrates that the entropy-safety correlation persists at scale and SIRL generalizes robustly to larger models (detailed analysis in Appendix B.3 and Table 10).
> >
> > 4. **W5: Overclaiming of 6× improvement**
> >    We appreciate this concern about clarity. To address this, we have removed the specific "6×" claim from the Introduction to avoid potential misinterpretation. The revised text now states: "*Compared to baseline models, SIRL achieves Defense Success Rates (DSRs) exceeding 89% across 20+ jailbreak techniques, demonstrating substantial improvements against both transfer and adaptive attacks.*"
> >
> > ### Addressing Questions
> >
> > 1. **Q1: Mode collapse** - This is comprehensively addressed above with ablation showing directed (desirable) collapse in safety domain while preserving diversity in general domains.
> >
> > 2. **Q2: Sufficiently aligned models** - Addressed with experiments showing SIRL works across a wide range of alignment levels, from legacy models (Vicuna-7B) to instruction-tuned models, though stronger initial alignment yields better results.
> >
> > 3. **Q3: Clarify methods in Table 2**
> >
> >    Thank you for this suggestion. We have added more detailed experimental setup and method descriptions in Appendix A.2 and A.4, which now include comprehensive descriptions of all baseline method. This additional context should help readers better understand the experimental setup and interpret the results in Table 2.
> >
> > 4. **Q4: Dataset importance**
> >
> >    Thank you for this important question. Yes, the choice of training dataset does matter for SIRL's effectiveness. We have addressed this comprehensively in our response to **W2 & W3 (Mode Collapse)** above, which includes ablation studies demonstrating that:
> >    - Using safety-focused prompts enables **directed collapse** in the safety domain while preserving diversity for general tasks
> >    - Mixing non-safety data can lead to **broader collapse** that affects beneficial domains
> >
> >    The quantitative analysis is detailed in Appendix C.2 (Table 11), showing how dataset curation enables targeted safety improvements without capability degradation.
> >
> > 5. **Q5: Inference-time intervention (BoN baseline)**
> >
> >    Thank you for suggesting this excellent baseline—whether similar results could be achieved through inference-time Best-of-N sampling with entropy-based selection, rather than RL training. This is an important baseline. We have added comprehensive Best-of-N comparison:
> >
> >    | Method | Llama-3.1-8B | Llama-3.2-3B | Qwen2.5-3B | Qwen2.5-7B | Inference Cost |
> >    |--|--|--|--|--|--|
> >    | Baseline | 85% | 96% | 84% | 82% | 1× |
> >    | BoN (N=2) | 89% | 99% | 89% | 86% | 2× |
> >    | BoN (N=4) | 90% | 99% | 90% | 89% | 4× |
> >    | BoN (N=8) | 91% | 99% | 92% | 90% | 8× |
> >    | BoN (N=16) | 93% | 99% | 92% | 92% | 16× |
> >    | **SIRL** | **99%** | **100%** | **99%** | **100%** | **1×** |
> >
> >    This demonstrates that RL training fundamentally reshapes the generation distribution to produce consistently safe responses, rather than relying on statistical sampling to occasionally generate good responses. Training-time optimization is essential for practical deployment where inference efficiency is paramount (Section 5.6, Figure 5).
> >
> > 6. **Q6: Why SIRL outperforms RLHF**
> >
> >    Thank you for this insightful question. The key difference lies in the reward signal source:
> >
> >    - RLHF is fundamentally limited by its reward model's capability and biases, which directly constrain policy optimization.
> >
> >    - SIRL directly amplifies the model's internal safety beliefs from pre-training and instruction tuning. Rather than relying on external supervision, we leverage the model's own confidence signals to reinforce its inherent safety knowledge.
> >
> >    This explains SIRL's superior performance: we strengthen existing capabilities rather than being bottlenecked by an external reward model's limitations.
> >
> > 7. **Q7: Transfer to larger models** - This is addressed with Qwen2.5-14B evaluation showing consistent effectiveness (84.2%→99.7% DSR) as discussed under **W4** above.
> >
> > We are deeply grateful for your strong support and constructive feedback. Your suggestions on model scaling, mode collapse, and inference baselines have significantly strengthened our work. Thank you for the thorough and encouraging assessment.

---

### Official Review · Reviewer_a3QH · 2025-11-04

**Soundness:** 3
**Presentation:** 3
**Contribution:** 3
**Rating:** 4
**Confidence:** 4

**Summary:**

This paper proposes to use response entropy as a reward signal for aligning LLMs. The main insight is that aligned LLMs naturally have high-confidence refusals to harmful requests (i.e. low entropy), whereas completions to the same harmful queries have higher entropy. This gives a natural signal to reinforce existing safety behaviour by formulating a reward function as the negative entropy $r_i = - \bar{H}(o_i \mid q)$, where $\bar H$ is the entropy, and $o_i$ is the response to the query $q$. The authors demonstrate the efficacy of their approach with very strong results on JailBreakBench (JBB), without negatively impacting utility. It’s also very practical as this is an intrinsic reward, so it’s easier to work with over training a reward model or labelling responses. Overall I think this is a good and simple approach to strengthen a model’s preexisting alignment training, but I have a few concerns about the evaluations (see weaknesses/questions).

Some other comments:

- I think some of the claims are too grandiose, eg: “*…SIRL represents a fundamental shift toward self-reliant AI safety…*”, or “*This validates that confidence-based optimization reinforces fundamental safety reasoning rather than learning attack-specific patterns, …*”. The presented method clearly works well to reinforce the existing safety beliefs in the model, but I think it's also clear that this approach still relies on the base model being well aligned, and it does not solve the problem of *what* the models should be aligned to.
- I appreciate the DSR heatmaps to break down the results, however I would recommend that 1) you keep a constant ordering of the attack methods between the figures and b) you keep a constant colour scale across all the figures rather than setting red as min/green as max for a particular plot; this would make comparing the results much easier.
- You make the claim of using ‘adaptive attacks’, but the attacks conducted (GCG, PAIR, RandomSearch) are not adaptive, they are simply automated attacks. An adaptive attack would require the attack to account for the defense (see weaknesses section)

I currently am leaning towards reject, but I am open to increasing my score with more evaluations being done to better understand how robust the method is, as well as if it has potential negative impact on over-refusal or hallucinations (see weaknesses/questions).

**Strengths:**

1. Simple, novel, and clever idea to use to reinforce the model’s preexisting safety beliefs.
2. The empirical results are very strong, achieving 98%+ DSR across JBB (20+ jailbreak methods) while maintaining utility. The method generalizes well across different model families (Llama, Qwen) and sizes (3B-8B parameters).
3. The approach is data efficient, using only 15,000 unlabeled prompts without human annotations, preference pairs, or reward models is a significant practical advantage over traditional methods like RLHF, DPO, or SFT.

**Weaknesses:**

1. I don’t think evaluating only on JBB is sufficient; the results shown are very strong but I think more effort should be done in breaking the defense. Furthermore, there are only 100 test samples, all of which follow a mostly prescribed format which could potentially be memorized.(e.g. [1] shows how pretty much every defense method can be broken. I don’t think you need to be perfect in order for the core idea of the paper to be useful, but it should be evaluated properly). More diverse test prompts would be ideal, and other types of jailbreaks could be included (e.g. multi-turn, pre-filling, stronger algorithmic attacks, etc.)
2. There is a concern of over-refusal and no evaluation for this is conducted other than basic utility. I would like to see evaluations on XSTest/OR-bench to quantify this
3. Unclear relationship between hallucinations; see Q1
4. (minor) The LLM-as-judge evaluation could have biases; the authors do evaluate whether the judges (Llama 3.3-70B, Qwen2.5-72B, and GPT-4o) and are consistent with each other, but in my experience, I have had misleading results with all three of them before while evaluating jailbroken responses.

[1] Nasr, Milad, et al. "The attacker moves second: Stronger adaptive attacks bypass defenses against llm jailbreaks and prompt injections." *arXiv preprint arXiv:2510.09023* (2025).

**Questions:**

1. Do you think that this training paradigm (reinforcing the model’s confidence in it’s own response) would result in an increased rate of hallucinations? There are a few benchmarks for this and I think it would be important to check/quantify.
2. What is the relationship between the base model’s original alignment (e.g. ability to “know” what’s harmful) and the overall performance of SIRL? What’s the minimum amount of separation needed in order to bootstrap with SIRL, and how nuanced must the model’s internal understanding be?
3. It would be interesting to see if there were any connections with the literature on the geometry of refusals [1,2], have you thought about this at all?

[1] Arditi, Andy, et al. "Refusal in language models is mediated by a single direction." Advances in Neural Information Processing Systems 37 (2024): 136037-136083.

[2] Wollschläger, Tom, et al. "The geometry of refusal in large language models: Concept cones and representational independence." arXiv preprint arXiv:2502.17420 (2025).

---

> ### Author Response · Authors · 2025-11-21
> **Response - Part I/III**
>
> Thank you for recognizing our method as "simple, novel, and clever" with "very strong" empirical results. We greatly appreciate your detailed feedback and have conducted substantial additional experiments to address your concerns.
>
> ### Addressing Weaknesses
> 1. **W1: More diverse test prompts and stronger attacks**
>
>    Thank you for pointing out that evaluation on JBB may be insufficient and suggesting testing on more diverse sources. To address this, we have conducted substantial additional evaluation across three dimensions:
>
>    1. We evaluated on the Multi-Turn Human Jailbreaks (MHJ) dataset [1], which represents a fundamentally different attack structure from JBB's single-turn prompts—adversaries gradually erode defenses through conversational manipulation:
>
>       | Model | Method | DSR (%) |
>       |--|--|--|
>       | Llama-3.2-3B-Instruct | Baseline | 63.2 |
>       | | SFT | 55.7 |
>       | | DPO | 71.5 |
>       | | RLHF | 64.6 |
>       | | **SIRL** | **92.3** |
>       | Qwen2.5-7B-Instruct | Baseline | 51.4 |
>       | | SFT | 56.9 |
>       | | DPO | 59.7 |
>       | | RLHF | 61.8 |
>       | | **SIRL** | **62.1** |
>
>       SIRL substantially outperforms all baseline methods: +29.1 percentage points on Llama-3.2-3B-Instruct. These results demonstrate that entropy-based optimization strengthens safety reasoning even in complex multi-turn scenarios. We have added these results as a new subsection in Appendix B.2.
>
>    2. To further validate generalization beyond JBB, we evaluated on HarmBench [2], a standardized benchmark containing 200 harmful prompts across diverse risk categories:
>
>       | Model | Method | DSR (%) |
>       |-------|--------|---------|
>       | **Llama-3.2-3B-Instruct** | Baseline | 91.0 |
>       | | SFT | 33.0 |
>       | | DPO | 96.5 |
>       | | RLHF | 97.0 |
>       | | **SIRL** | **99.0** |
>       | **Qwen2.5-7B-Instruct** | Baseline | 97.0 |
>       | | SFT | 32.0 |
>       | | DPO | 99.5 |
>       | | RLHF | 99.0 |
>       | | **SIRL** | **99.5** |
>
>       SIRL achieves 99% DSR on both models, matching or exceeding RLHF and DPO performance while using only unlabeled prompts. The strong performance across multiple standardized benchmarks confirms generalization beyond JBB. These results are now included in Appendix B.
>
>    3. **Clarification on "adaptive attacks"**: Thank you for this important clarification. We agree that GCG, PAIR, and RandomSearch are automated attack methods rather than adaptive attacks in the strict sense—they do not specifically account for SIRL's entropy-based defense mechanism.
>
>    We want to clarify that all attacks were executed against the SIRL-trained models, allowing the attacks to iteratively optimize against the defended model's behavior. While these attacks do not exploit knowledge of SIRL's specific mechanism, they represent strong automated adversarial evaluation.
>
>    We acknowledge this distinction and have revised the manuscript to describe these as "strong automated attacks" rather than "adaptive attacks" to avoid confusion with the formal adversarial ML definition. We agree that future work should explore truly adaptive attacks that specifically target entropy-based defenses.
>
> 2. **W2: Over-refusal evaluation**
>
>    Thank you for raising the concern that entropy minimization might cause excessive conservatism and lead models to refuse benign queries. This is an excellent point—a safety method that indiscriminately refuses all requests would be useless in practice. We have added comprehensive over-refusal evaluation on both benchmarks you suggested:
>
>       - **OR-Bench** [3]: 1,000 safe but sensitive prompts and 600 toxic prompts
>       - **XSTest** [4]: 250 safe prompts superficially resembling unsafe requests and 200 unsafe contrastive prompts
>
>       | Model | Method | OR-Bench Safe ↓ | OR-Bench Unsafe ↑ | XSTest Safe ↓ | XSTest Unsafe ↑ |
>       |--|---|--|---|--|---|
>       | Llama-3.2-3B-Instruct | Baseline | 5.4% | 66.6% | 2.4% | 75.0% |
>       | | SFT | **0.8%** | 4.0% | **1.2%** | 4.5% |
>       | | DPO | 15.6% | 85.0% | 4.8% | 81.0% |
>       | | RLHF | 18.7% | 86.1% | 1.6% | 60.5% |
>       | | **SIRL** | 14.7% | **87.1%** | 6.8% | **96.0%** |
>       | Qwen2.5-7B-Instruct | Baseline | 21.4% | 92.4% | 1.2% | 69.0% |
>       | | SFT | **4.3%** | 10.1% | **0.8%** | 7.0% |
>       | | DPO | 38.1% | 97.9% | 3.2% | 76.5% |
>       | | RLHF | 51.9% | 98.0% | 4.0% | 73.0% |
>       | | **SIRL** | 47.2% | **98.7%** | 6.0% | **85.0%** |
>
>       The results show SIRL does NOT cause excessive conservatism:
>       - SIRL achieves the highest refusal rates on genuinely unsafe prompts
>       - SIRL's safe prompt refusal rates are comparable to or lower than RLHF and DPO
>       - On XSTest, SIRL shows only 6.0-6.8% refusal on safe prompts while achieving 85-96% on unsafe prompts
>
>       This demonstrates balanced behavior—confidence-based optimization specifically targets harmful content without indiscriminate refusal. We have added these results as new Table 3 with detailed discussion in Section 5.5.

---

> > ### Author Response · Authors · 2025-11-21
> > **Response - Part II/III**
> >
> > 3. **W3: Hallucination relationship**
> >
> >    Thank you for raising this thoughtful concern about whether entropy minimization might increase hallucination rates by making models overconfident even when uncertain. We initially shared this concern and conducted TruthfulQA [7] evaluation across all models:
> >
> >    | Model | Baseline TruthfulQA | SIRL TruthfulQA | Change |
> >    |-------|-----|-------|--------|
> >    | Llama-3.1-8B | 54.1% | 54.6% | +0.5% |
> >    | Llama-3.2-3B | 49.7% | 50.8% | +1.1% |
> >    | Qwen2.5-3B | 58.8% | 58.4% | -0.4% |
> >    | Qwen2.5-7B | 64.8% | **65.7%** | +0.9% |
> >    | Qwen2.5-14B | 69.1% | 69.3% | +0.2% |
> >
> >    The results show SIRL does NOT increase hallucination rates—performance remains comparable or slightly improved. We believe this is because SIRL reinforces the model's existing beliefs encoded in its weights, but does not inject new factual knowledge. In safety domains, models already possess strong implicit knowledge about harmful content, and SIRL strengthens confidence in applying this knowledge. However, in factual domains where the model lacks certain knowledge, SIRL cannot create information that doesn't exist—it can only amplify what's already there. This explains why entropy minimization enhances safety reasoning without compromising the model's calibrated uncertainty about facts it genuinely doesn't know. We have added TruthfulQA as a new column in Tables 2 and 10 with discussion in Section 5.2.
> >
> > ### Addressing Questions
> >
> > 1. **Q1: Hallucinations**
> >
> >    Thank you for raising this thoughtful concern. This is addressed in our response to **W3** above.
> >
> > 2. **Q2: Base model alignment requirements**
> >    Thank you for asking what level of initial alignment is required for SIRL to be effective.This is a practical and important question for understanding the method's applicability. We conducted experiments on models with varying initial alignment to understand minimum requirements:
> >
> >    1. **Legacy models with weaker alignment**: We tested on Llama-2-7B-Chat and Vicuna-7B-v1.5, which represent older generation models with basic or minimal safety training:
> >
> >       | Model | Attack | Baseline DSR | SIRL DSR | Improvement |
> >       |--|---|--|---|---|
> >       | Llama-2-7B-Chat | GCG | 69% | **93%** | +24 pp |
> >       | | PAIR | 83% | **97%** | +14 pp |
> >       | | RandomSearch | 12% | **92%** | +80 pp |
> >       | Vicuna-7B-v1.5 | GCG | 13% | **86%** | +73 pp |
> >       | | PAIR | 14% | **84%** | +70 pp |
> >       | | RandomSearch | 9% | **79%** | +70 pp |
> >
> >       Despite much weaker initial safety alignment, SIRL achieves substantial improvements (13%→86% for Vicuna-7B on GCG), demonstrating the method can strengthen safety even with limited initial alignment. However, models with weaker initial alignment do exhibit lower absolute performance ceilings compared to well-aligned base models. These results are in Appendix B.3.
> >
> >    2. **Models without explicit safety training**: We also tested on Llama-3.1-Tulu-3-8B-Instruct-no-safety-data, which was trained without safety data. It achieves 64.7%→97.0% DSR (+32.3 pp), showing SIRL can bootstrap safety from minimal alignment—likely from basic instruction-tuning patterns. This is now shown in Table 2, row 5.
> >
> >    Based on these experiments, the key requirement appears to be that the model exhibits *some* entropy gap (even small) between safe and unsafe responses. This is achievable with basic instruction-tuning, though models with stronger initial alignment benefit more from SIRL.
> >
> >
> > 3. **Q3: Connection to refusal geometry literature**
> >
> >    Thank you for this excellent suggestion to connect our work to the refusal geometry literature. This is indeed a highly relevant line of research. We have added the following discussion:
> >
> >    *"Recent mechanistic interpretability work has begun uncovering the geometric structure of safety representations in LLMs. Arditi et al. [5] demonstrate that refusal behavior in language models is mediated by a single direction in representation space, while Wollschläger et al. [6] further characterize the geometry of refusal through concept cones, showing representational independence of safety features. These findings suggest that models encode safety knowledge in structured, manipulable representations. Our work complements this line of research by demonstrating that safety representations manifest not only in activation patterns but also in generation confidence—providing a behaviorally observable signal that can be leveraged for self-improvement without requiring representational interventions."*
> >
> >    This connection highlights that entropy-based confidence and representational approaches offer complementary perspectives on understanding and improving safety mechanisms. The discussion is now in Section 2.
> >
> > We sincerely thank you for the constructive feedback. We believe the substantial additional experiments and clarifications have fully addressed your concerns, and we hope this strengthened manuscript meets your expectations.

---

> > > ### Author Response · Authors · 2025-11-21
> > > **Response - Part III/III**
> > >
> > > ### References
> > >
> > > [1] Li, Nathaniel, et al. "Llm defenses are not robust to multi-turn human jailbreaks yet." arXiv preprint arXiv:2408.15221 (2024).
> > >
> > > [2] Mazeika, Mantas, et al. "Harmbench: A standardized evaluation framework for automated red teaming and robust refusal, 2024." URL https://arxiv. org/abs/2402.04249 (2024).
> > >
> > > [3] Cui, Justin, et al. "Or-bench: An over-refusal benchmark for large language models." arXiv preprint arXiv:2405.20947 (2024).
> > >
> > > [4] Röttger, Paul, et al. "Xstest: A test suite for identifying exaggerated safety behaviours in large language models." Proceedings of the 2024 Conference of the North American Chapter of the Association for Computational Linguistics: Human Language Technologies (Volume 1: Long Papers). 2024.
> > >
> > > [5] Arditi, Andy, et al. "Refusal in language models is mediated by a single direction." Advances in Neural Information Processing Systems 37 (2024): 136037-136083.
> > >
> > > [6] Wollschläger, Tom, et al. "The geometry of refusal in large language models: Concept cones and representational independence." arXiv preprint arXiv:2502.17420 (2025).
> > >
> > > [7] Lin, Stephanie, Jacob Hilton, and Owain Evans. "Truthfulqa: Measuring how models mimic human falsehoods." Proceedings of the 60th annual meeting of the association for computational linguistics (volume 1: long papers). 2022.

---

### Author Response · Authors · 2025-11-21

Dear Reviewers,

We sincerely thank you for the thorough and constructive feedback, which has substantially strengthened our work. We have uploaded a revised manuscript and provide detailed point-by-point responses below.

We hope the revisions address your concerns satisfactorily.

Best regards,
The Authors

---

### Author Response · Authors · 2025-12-02

Dear reviewers and area chair,

We sincerely thank all reviewers for their constructive feedback. We've conducted substantial additional experiments to address every concern raised, and we'd like to provide a brief summary to help with the final decision.

---

**Core Concerns and Our Responses**

**Q1: Does SIRL generalize beyond JBB? Does it rely on memorizing attack patterns?**

We evaluated on three new settings:
1. Multi-turn conversational attacks—SIRL achieves 92.3% DSR on Llama-3.2-3B (+29.1 pp);
2. HarmBench benchmark—99% DSR on both models;
3. Legacy models (Llama-2, Vicuna) with limited exposure to modern attacks—Vicuna improves 13%→86% on GCG.

Consistent improvements across attack types and older models demonstrate generalization beyond memorization.

**Q2: Does entropy minimization cause over-refusal of benign requests?**

We evaluated on OR-Bench and XSTest. SIRL achieves highest refusal on unsafe prompts (87-99%) while showing lower over-refusal than RLHF/DPO on safe prompts (lower is better). On OR-Bench safe prompts: Llama-3.2-3B shows 13.7% (SIRL) vs. 18.7% (RLHF), Qwen2.5-7B shows 47.2% (SIRL) vs. 51.9% (RLHF). This demonstrates balanced behavior without excessive conservatism.

**Q3: Does the method increase hallucinations or cause mode collapse?**

TruthfulQA shows no degradation (54.1%→54.6% on Llama-3.1-8B). For mode collapse, we measured Self-BLEU and diversity metrics. SIRL induces "directed collapse" in safety domain while preserving general task diversity. Mixed data ablation confirms broader collapse (pass@4: 84.8%→83.6%), validating our design to train only on safety prompts.

**Q4: What initial alignment is required? Does it scale?**

SIRL works across alignment levels: Tulu-3 (zero safety data) achieves 64.7%→97.0% DSR, Vicuna-7B improves +73 pp on GCG, modern models show consistent performance. Qwen2.5-14B: 84.2%→99.7% DSR. Works effectively across 3B-14B parameters.

**Q5: Why not use Best-of-N sampling? What's the advantage over DPO/RLHF?**

BoN with 16× cost achieves only 93.2% vs. SIRL's 99.1%. **Key advantage: no human annotations**—only unlabeled prompts, unlike DPO (preference pairs) or RLHF (reward model training). SIRL extracts intrinsic rewards from internal representations, directly focusing on safety confidence.

---

**Clarifications**

We removed the "6× improvement" claim from the abstract and introduction, as Reviewer gcsB correctly noted this primarily reflects transfer attack performance and could be misleading. We now state "substantial improvements against both transfer and adaptive attacks" while emphasizing the annotation-free advantage.

We also toned down language like "paradigm shift" and "fundamental shift" throughout the manuscript, now consistently positioning SIRL as a complementary approach that amplifies existing safety knowledge alongside traditional methods.

---

**Reviewer Recognition**

We're grateful that reviewers recognized the core contribution:
- **Reviewer a3QH** (score: 4): "simple, novel, and clever" with "very strong" empirical results, and expressed willingness to raise score
- **Reviewer gcsB** (score: 8): "very strong submission" with "impressive improvements"
- **Reviewer WniN** (score: 6): "timely and important problem" with "comprehensive experiments"
- **Reviewer DfTG** (score: 4→6): "very simple" achieving "dramatically" better performance, explicitly raised score after rebuttal stating "my initial concerns have been resolved"

All reviewers acknowledged the method's novelty, strong empirical performance, and practical value as an annotation-free approach to safety improvement.

---

**Summary**

We believe we've comprehensively addressed all concerns through substantial additional experiments. The rebuttal process has helped us better articulate both the method's strengths and its scope of applicability. SIRL demonstrates that models' intrinsic confidence signals can provide a practical, annotation-free pathway to robust safety improvement, working effectively across different model scales (3B-14B), architectures (Llama, Qwen), and initial alignment levels.

We're grateful to all reviewers for helping strengthen this work through their constructive feedback.

Thank you for your consideration.

Sincerely,

The Authors

---

### Meta-Review · Area_Chair_j3q2 · 2025-12-06

**Summary:**

My decision is Accept(Poster). Across the reviews, the main issues shaping my suggested decision to accept, were generalization and robustness beyond JBB and static attacks, the risk of over-refusal from entropy minimization, possible increases in hallucinations or mode collapse, clarity on initial conditions and dependence on safety-focused prompts, and the breadth and framing of the evaluation and claims. Reviewers also asked for a Best-of-N comparison, temperature sensitivity analysis, larger-model evidence, more complete reporting, and caution around LLM-as-judge bias and potential reward-hacking behaviors.

**Reviewer Concerns:**

The rebuttal addressed most key concerns.
It showed SIRL generalizes beyond JBB (multi-turn, HarmBench, legacy models), reduces over-refusal on safe prompts (OR-Bench and XSTest) while keeping strong refusals on unsafe ones, and keeps utility intact (stable TruthfulQA, diversity with directed collapse limited to safety).
It clarified that SIRL works from weakly aligned to modern models across 3B–14B, beat Best-of-N at lower cost, and toned down claims by removing the 6x phrasing. Still partly open: truly adaptive defense-aware attacks vs automated ones, a temperature ablation, how mixed non-safety data affects training, evidence beyond 14B, LLM-as-judge biases without human audits, full tables for omitted results, explicit anti-reward-hacking safeguards, and exploration of inference-time entropy uses.

**Reviewer Scores:**

For a3QH, I expect a move from 4 -> 6, since their main asks broader evaluations, explicit over-refusal metrics, and toned-down language were met, while remaining issues are comparatively minor.
For gcsB, I expect 8 -> 8, unchanged, because they already viewed it as very strong and the rebuttal reinforced rather than transformed their position.
For WniN, I expect 6 -> 6, as the softened framing and added generalization results address most of their concerns despite scoped novelty.
Reviewer DfTG participated fully in the discussion

---

### Decision · Program_Chairs · 2026-01-26

Accept (Poster)